# D-FINE: Redefine Regression Task in DETRs as Fine-grained Distribution Refinement

**Yansong Peng[1], Hebei Li[1], Peixi Wu[1], Yueyi Zhang[1]***, **Xiaoyan Sun[1, 2]***, **Feng Wu[1, 2]**

[1]University of Science and Technology of China

[2]Institute of Artificial Intelligence, Hefei Comprehensive National Science Center

{pengyansong, lihebei, wupeixi}@mail.ustc.edu.cn

{zhyuey, sunxiaoyan, fengwu}@ustc.edu.cn

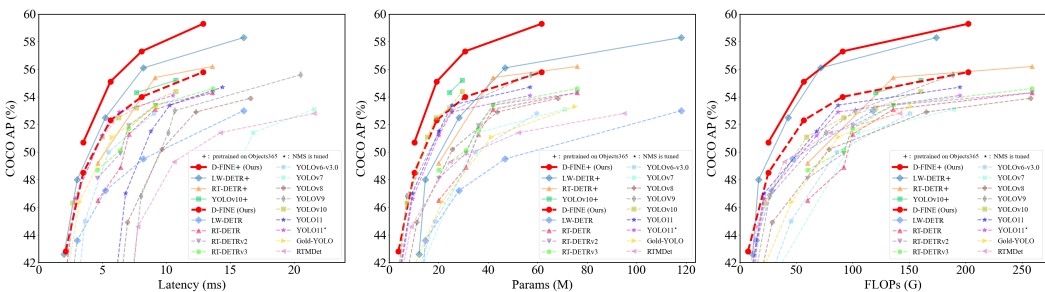

Figure 1: Comparisons with other detectors in terms of latency (left), model size (mid), and computational cost (right). We measure end-to-end latency using TensorRT FP16 on an NVIDIA T4 GPU.

## Abstract

We introduce **D-FINE**, a powerful real-time object detector that achieves outstanding localization precision by redefining the bounding box regression task in DETR models. D-FINE comprises two key components: **Fine-grained Distribution Refinement (FDR)** and **Global Optimal Localization Self-Distillation (GO-LSD)**. FDR transforms the regression process from predicting fixed coordinates to iteratively refining probability distributions, providing a fine-grained intermediate representation that significantly enhances localization accuracy. GO-LSD is a bidirectional optimization strategy that transfers localization knowledge from refined distributions to shallower layers through self-distillation, while also simplifying the residual prediction tasks for deeper layers. Additionally, D-FINE incorporates lightweight optimizations in computationally intensive modules and operations, achieving a better balance between speed and accuracy. Specifically, D-FINE-L / X achieves 54.0% / 55.8% AP on the COCO dataset at 124 / 78 FPS on an NVIDIA T4 GPU. When pretrained on Objects365, D-FINE-L / X attains 57.1% / 59.3% AP, surpassing all existing real-time detectors. Furthermore, our method significantly enhances the performance of a wide range of DETR models by up to 5.3% AP with negligible extra parameters and training costs. Our code and models: https://github.com/Peterande/D-FINE.

## 1 Introduction

The demand for real-time object detection has been increasing across various applications (Arani et al., 2022). Among the most influential real-time detectors are the YOLO series (Redmon et al., 2016a; Wang et al., 2023a;b; Glenn., 2023; Wang & Liao, 2024; Wang et al., 2024a; Glenn., 2024), widely recognized for their efficiency and robust community ecosystem. As a strong competitor, the Detection Transformer (DETR) (Carion et al., 2020; Zhu et al., 2020; Liu et al., 2021; Li et al., 2022; Zhang et al., 2022) offers distinct advantages due to its transformer-based architecture, which allows

---

*Corresponding authors

for global context modeling and direct set prediction without reliance on Non-Maximum Suppression (NMS) and anchor boxes. However, they are often hindered by high latency and computational demands (Zhu et al., 2020; Liu et al., 2021; Li et al., 2022; Zhang et al., 2022). RT-DETR (Zhao et al., 2024) addresses these limitations by developing a real-time variant, offering an end-to-end alternative to YOLO detectors. Moreover, LW-DETR (Chen et al., 2024) has shown that DETR can achieve higher performance ceilings than YOLO, especially when trained on large-scale datasets like Objects365 (Shao et al., 2019).

Despite the substantial progress made in real-time object detection, there remain potential aspects for improvement in detector performance. One key aspect is the formulation of bounding box regression. Most detectors predict bounding boxes by regressing fixed coordinates, treating edges as precise values modeled by Dirac delta distributions (Liu et al., 2016; Ren et al., 2015; Tian et al., 2019; Lyu et al., 2022). While this approach has shown significant success in maintaining efficiency, using fixed coordinates inherently comes with limitations in modeling localization uncertainty. Consequently, models are constrained to use L1 loss and IoU loss, which provide insufficient guidance for adjusting each edge independently (Girshick, 2015). This makes the optimization process sensitive to small coordinate changes, potentially leading to slow convergence and suboptimal performance. Although methods like GFocal (Li et al., 2020; 2021) address uncertainty through probability distributions, they remain limited by anchor dependency, coarse localization, and lack of iterative refinement. Another aspect lies in maximizing the efficiency of real-time detectors, which are constrained by limited computation and parameter budgets to maintain speed. Knowledge distillation (KD) is a promising solution, transferring knowledge from larger teachers to smaller students to improve performance without increasing costs (Hinton et al., 2015). However, traditional KD methods like Logit Mimicking and Feature Imitation have proven inefficient for detection tasks and can even cause performance drops in state-of-the-art models (Zheng et al., 2022). In contrast, localization distillation (LD) has shown better results for detection. Nevertheless, integrating LD remains challenging due to its substantial training overhead and incompatibility with anchor-free detectors.

To address these issues, we propose **D-FINE**, a novel real-time object detector that redefines bounding box regression and introduces an effective self-distillation strategy. Our approach tackles the problems of difficult optimization in fixed-coordinate regression, the inability to model localization uncertainty, and the need for effective distillation with less training cost. We introduce **Fine-grained Distribution Refinement (FDR)** to transform bounding box regression from predicting fixed coordinates to modeling probability distributions, providing a more fine-grained intermediate representation. FDR refines these distributions iteratively in a residual manner, allowing for progressively finer adjustments and improving localization precision. Recognizing that deeper layers produce more accurate predictions by capturing richer localization information within their probability distributions, we introduce **Global Optimal Localization Self-Distillation (GO-LSD)**. GO-LSD transfers localization knowledge from deeper layers to shallower ones with negligible extra training cost. By aligning shallower layers' predictions with refined outputs from later layers, the model learns to produce better early adjustments, accelerating convergence and improving overall performance. Furthermore, we streamline computationally intensive modules and operations in existing real-time DETR architectures (Zhao et al., 2024; Chen et al., 2024), making D-FINE faster and more lightweight. While such modifications typically result in performance loss, FDR and GO-LSD effectively mitigate this degradation, achieving a better balance between speed and accuracy.

Experimental results on the COCO dataset (Lin et al., 2014a) demonstrate that D-FINE achieves state-of-the-art performance in real-time object detection, surpassing existing models in accuracy and efficiency. D-FINE-L and D-FINE-X achieve 54.0% and 55.8% AP, respectively on COCO `val2017`, running at 124 FPS and 78 FPS on an NVIDIA T4 GPU. After pretraining on larger datasets like Objects365 (Shao et al., 2019), the D-FINE series attains up to 59.3% AP, surpassing all existing real-time detectors, showcasing both scalability and robustness. Moreover, our method enhances a variety of DETR models by up to 5.3% AP with negligible extra parameters and training costs, demonstrating its flexibility and generalizability. In conclusion, D-FINE pushes the performance boundaries of real-time detectors. By addressing key challenges in bounding box regression and distillation efficiency through FDR and GO-LSD, we offer a meaningful step forward in object detection, inspiring further exploration in the field.

## 2 RELATED WORK

**Real-Time / End-to-End Object Detectors.** The YOLO series has led the way in real-time object detection, evolving through innovations in architecture, data augmentation, and training techniques (Redmon et al., 2016a; Wang et al., 2023a;b; Glenn., 2023; Wang & Liao, 2024; Glenn., 2024). While efficient, YOLOs typically rely on Non-Maximum Suppression (NMS), which introduces latency and instability between speed and accuracy. DETR (Carion et al., 2020) revolutionizes object detection by removing the need for hand-crafted components like NMS and anchors. Traditional DETRs (Zhu et al., 2020; Meng et al., 2021; Zhang et al., 2022; Wang et al., 2022; Liu et al., 2021; Li et al., 2022; Chen et al., 2022a;c) have achieved excelling performance but at the cost of high computational demands, making them unsuitable for real-time applications. Recently, RT-DETR (Zhao et al., 2024) and LW-DETR (Chen et al., 2024) have successfully adapted DETR for real-time use. Concurrently, YOLOv10 (Wang et al., 2024a) also eliminates the need for NMS, marking a significant shift towards end-to-end detection within the YOLO series.

**Distribution-Based Object Detection.** Traditional bounding box regression methods (Redmon et al., 2016b; Liu et al., 2016; Ren et al., 2015) rely on Dirac delta distributions, treating bounding box edges as precise and fixed, which makes modeling localization uncertainty challenging. To address this, recent models have employed Gaussian or discrete distributions to represent bounding boxes (Choi et al., 2019; Li et al., 2020; Qiu et al., 2020; Li et al., 2021), enhancing the modeling of uncertainty. However, these methods all rely on anchor-based frameworks, which limits their compatibility with modern anchor-free detectors like YOLOX (Ge et al., 2021) and DETR (Carion et al., 2020). Furthermore, their distribution representations are often formulated in a coarse-grained manner and lack effective refinement, hindering their ability to achieve more accurate predictions.

**Knowledge Distillation.** Knowledge distillation (KD) (Hinton et al., 2015) is a powerful model compression technique. Traditional KD typically focuses on transferring knowledge through Logit Mimicking (Zagoruyko & Komodakis, 2017; Mirzadeh et al., 2020; Son et al., 2021). Fit-Nets (Romero et al., 2015) initially propose Feature Imitation, which has inspired a series of subsequent works that further expand upon this idea (Chen et al., 2017; Dai et al., 2021; Guo et al., 2021; Li et al., 2017; Wang et al., 2019). Most approaches for DETR (Chang et al., 2023; Wang et al., 2024b) incorporate hybrid distillations of both logit and various intermediate representations. Recently, localization distillation (LD) (Zheng et al., 2022) demonstrates that transferring localization knowledge is more effective for detection tasks. Self-distillation (Zhang et al., 2019; 2021) is a special case of KD which enables earlier layers to learn from the model's own refined outputs, requiring far fewer additional training costs since there's no need to separately train a teacher model.

## 3 PRELIMINARIES

**Bounding box regression** in object detection has traditionally relied on modeling Dirac delta distributions, either using centroid-based $\{x, y, w, h\}$ or edge-distance $\{\mathbf{c}, \mathbf{d}\}$ forms, where the distances $\mathbf{d} = \{t, b, l, r\}$ are measured from the anchor point $\mathbf{c} = \{x_c, y_c\}$. However, the Dirac delta assumption, which treats bounding box edges as precise and fixed, makes it difficult to model localization uncertainty, especially in ambiguous cases. This rigid representation may limit optimization and potentially lead to localization errors with small prediction shifts.

To address these problems, GFocal (Li et al., 2020; 2021) regresses the distances from anchor points to the four edges using discretized probability distributions, offering a more flexible modeling of the bounding box. In practice, bounding box distances $\mathbf{d} = \{t, b, l, r\}$ is modeled as:

$$\mathbf{d} = d_{\max} \sum_{n=0}^{N} \frac{n}{N} \mathbf{P}(n), \tag{1}$$

where $d_{\max}$ is a scalar that limits the maximum distance from the anchor center, and $\mathbf{P}(n)$ denotes the probability of each candidate distance of four edges. While GFocal introduces a step forward in handling ambiguity and uncertainty through probability distributions, specific challenges in its regression approach persist: (1) *Anchor Dependency*: Regression is tied to the anchor box center, limiting prediction diversity and compatibility with anchor-free frameworks. (2) *No Iterative Refinement*: Predictions are made in one shot without iterative refinements, reducing regression robustness.

(3) *Coarse Localization*: Fixed distance ranges and uniform bin intervals can lead to coarse localization, especially for small objects, because each bin represents a wide range of possible values.

**Localization Distillation (LD)** is a promising approach, demonstrating that transferring localization knowledge is more effective for detection tasks (Zheng et al., 2022). Built upon GFocal, it enhances student models by distilling valuable localization knowledge from teacher models, rather than simply mimicking classification logits or feature maps. Despite its advantages, the method still relies on anchor-based architectures and incurs additional training costs.

## 4 METHOD

We propose **D-FINE**, a powerful real-time object detector that excels in speed, size, computational cost, and accuracy. D-FINE addresses the shortcomings of existing bounding box regression approaches by leveraging two key components: **Fine-grained Distribution Refinement (FDR)** and **Global Optimal Localization Self-Distillation (GO-LSD)**, which work in tandem to significantly enhance performance with negligible additional parameters and training time cost.

**(1) FDR** iteratively optimizes probability distributions that act as corrections to the bounding box predictions, providing a more fine-grained intermediate representation. This approach captures and optimizes the uncertainty of each edge independently. By leveraging the non-uniform weighting function, FDR allows for more precise and incremental adjustments at each decoder layer, improving localization accuracy and reducing prediction errors. FDR operates within an anchor-free, end-to-end framework, enabling a more flexible and robust optimization process.

**(2) GO-LSD** distill localization knowledge from refined distributions into shallower layers. As training progresses, the final layer produces increasingly precise soft labels. Shallower layers align their predictions with these labels through GO-LSD, leading to more accurate predictions. As early-stage predictions improve, the subsequent layers can focus on refining smaller residuals. This mutual reinforcement creates a synergistic effect, leading to progressively more accurate localization.

To further enhance the efficiency of D-FINE, we streamline computationally intensive modules and operations in existing real-time DETR architectures (Zhao et al., 2024), making D-FINE faster and more lightweight. Although these modifications typically result in some performance loss, FDR and GO-LSD effectively mitigate this degradation. The detailed modifications are listed in Table 4.

### 4.1 FINE-GRAINED DISTRIBUTION REFINEMENT

**Fine-grained Distribution Refinement (FDR)** iteratively optimizes a fine-grained distribution generated by the decoder layers, as shown in Figure 2. In the first decoder layer, preliminary bounding boxes are predicted by a traditional bounding box regression head, while the D-FINE head generates the initial probability distributions for the four edges (top, bottom, left, right). The preliminary bounding boxes serve as reference boxes, while subsequent layers focus on refining them by adjusting distributions in a residual manner. The refined distributions are then applied to adjust the four edges of the corresponding initial bounding box, progressively improving its accuracy with each iteration. Mathematically, let $\mathbf{b}^0 = \{x, y, w, h\}$ denote the initial bounding box prediction, where $\{x, y\}$ represents the predicted center of the bounding box, and $\{w, h\}$ represent the width and height of the box. We can then convert $\mathbf{b}^0$ into the center coordinates $\mathbf{c}^0 = \{x, y\}$ and the edge distances $\mathbf{d}^0 = \{t, b, l, r\}$, which represent the distances from the center to the top, bottom, left, and right edges. For the $l$-th layer, the refined edge distances $\mathbf{d}^l = \{t^l, b^l, l^l, r^l\}$ are computed as:

$$\mathbf{d}^l = \mathbf{d}^0 + \{h, h, w, w\} \cdot \sum_{n=0}^{N} W(n)\mathbf{Pr}^l(n), \quad l \in \{1, 2, \ldots, L\}, \tag{2}$$

where $\mathbf{Pr}^l(n) = \{\mathrm{Pr}_t^l(n), \mathrm{Pr}_b^l(n), \mathrm{Pr}_l^l(n), \mathrm{Pr}_r^l(n)\}$ represents four separate distributions, one for each edge. Each distribution predicts the likelihood of candidate offset values for the corresponding edge. These candidates are determined by the weighting function $W(n)$, where $n$ indexes the discrete bins out of $N$, with each bin corresponding to a potential edge offset. The weighted sum of the distributions produces the edge offsets. These edge offsets are then scaled by the height $h$ and width $w$ of the initial bounding box, ensuring the adjustments are proportional to the box size.

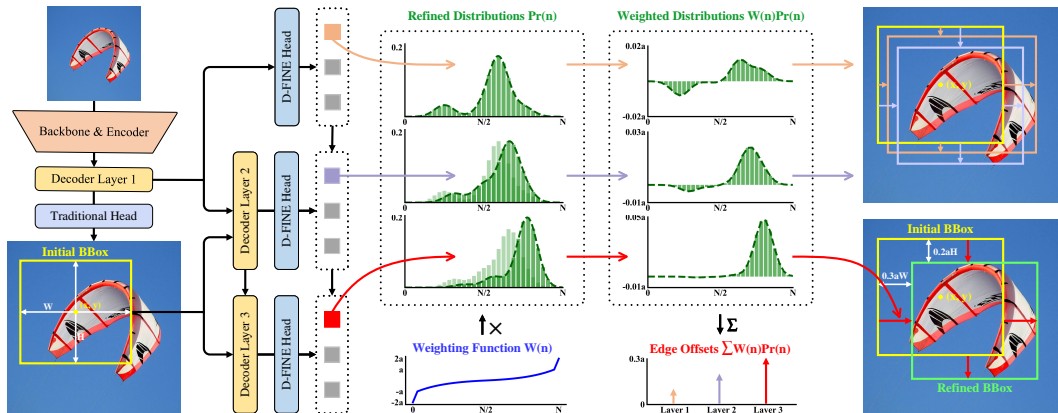

Figure 2: Overview of D-FINE with FDR. The probability distributions that act as a more fine-grained intermediate representation are iteratively refined by the decoder layers in a residual manner. Non-uniform weighting functions are applied to allow for finer localization.

The refined distributions are updated using residual adjustments, defined as follows:

$$\mathbf{Pr}^l(n) = \text{Softmax}\left(logits^l(n)\right) = \text{Softmax}\left(\Delta logits^l(n) + logits^{l-1}(n)\right), \quad (3)$$

where logits from the previous layer $logits^{l-1}(n)$ reflect the confidence in each bin's offset value for the four edges. The current layer predicts the residual logits $\Delta logits^l(n)$, which are added to the previous logits to form updated logits $logits^l(n)$. These updated logits are then normalized using the softmax function, producing the refined probability distributions.

To facilitate precise and flexible adjustments, the weighting function $W(n)$ is defined as:

$$W(n) = \begin{cases} 2 \cdot W(1) = -2a & n = 0 \\ c - c\left(\frac{a}{c} + 1\right)^{\frac{N-2n}{N-2}} & 1 \leq n < \frac{N}{2} \\ -c + c\left(\frac{a}{c} + 1\right)^{\frac{-N+2n}{N-2}} & \frac{N}{2} \leq n \leq N - 1 \\ 2 \cdot W(N-1) = 2a & n = N, \end{cases} \quad (4)$$

where $a$ and $c$ are hyper-parameters controlling the upper bounds and curvature of the function. As shown in Figure 2, the shape of $W(n)$ ensures that when bounding box prediction is near accurate, small curvature in $W(n)$ allows for finer adjustments. Conversely, if the bounding box prediction is far from accurate, the larger curvature near the edges and the sharp changes at the boundaries of $W(n)$ ensure sufficient flexibility for substantial corrections.

To further improve the accuracy of our distribution predictions and align them with ground truth values, inspired by Distribution Focal Loss (DFL) (Li et al., 2020), we propose a new loss function, Fine-Grained Localization (FGL) Loss, which is computed as:

$$\mathcal{L}_{\text{FGL}} = \sum_{l=1}^{L}\left(\sum_{k=1}^{K}\text{IoU}_k\left(\omega_\leftarrow \cdot \mathbf{CE}\left(\mathbf{Pr}^l(n)_k, n_\leftarrow\right) + \omega_\rightarrow \cdot \mathbf{CE}\left(\mathbf{Pr}^l(n)_k, n_\rightarrow\right)\right)\right)$$

$$\omega_\leftarrow = \frac{|\phi - W(n_\rightarrow)|}{|W(n_\leftarrow) - W(n_\rightarrow)|}, \quad \omega_\rightarrow = \frac{|\phi - W(n_\leftarrow)|}{|W(n_\leftarrow) - W(n_\rightarrow)|}, \quad (5)$$

where $\mathbf{Pr}^l(n)_k$ represents the probability distributions corresponding to the $k$-th prediction. $\phi$ is the relative offset calculated as $\phi = (\mathbf{d}^{GT} - \mathbf{d}^0)/\{h, h, w, w\}$. $\mathbf{d}^{GT}$ represents the ground truth edge-distance and $n_\leftarrow, n_\rightarrow$ are the bin indices adjacent to $\phi$. The cross-entropy (CE) losses with weights $\omega_\leftarrow$ and $\omega_\rightarrow$ ensure that the interpolation between bins aligns precisely with the ground truth offset. By incorporating IoU-based weighting, FGL loss encourages distributions with lower uncertainty to become more concentrated, resulting in more precise and reliable bounding box regression.

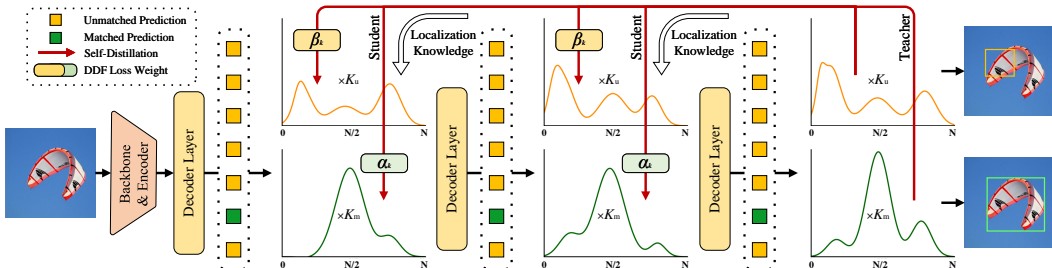

Figure 3: Overview of GO-LSD process. Localization knowledge from the final layer's refined distributions is distilled into shallower layers through DDF loss with decoupled weighting strategies.

## 4.2 GLOBAL OPTIMAL LOCALIZATION SELF-DISTILLATION

**Global Optimal Localization Self-Distillation** (GO-LSD) utilizes the final layer's refined distribution predictions to distill localization knowledge into the shallower layers, as shown in Figure 3. This process begins by applying Hungarian Matching (Carion et al., 2020) to the predictions from each layer, identifying the local bounding box matches at every stage of the model. To perform a global optimization, GO-LSD aggregates the matching indices from all layers into a unified union set. This union set combines the most accurate candidate predictions across layers, ensuring that they all benefit from the distillation process. In addition to refining the global matches, GO-LSD also optimizes unmatched predictions during training to improve overall stability, leading to improved overall performance. Although the localization is optimized through this union set, the classification task still follows a one-to-one matching principle, ensuring that there are no redundant boxes. This strict matching means that some predictions in the union set are well-localized but have low confidence scores. These low-confidence predictions often represent candidates with precise localization, which still need to be distilled effectively.

To address this, we introduce Decoupled Distillation Focal (DDF) Loss, which applies decoupled weighting strategies to ensure that high-IoU but low-confidence predictions are given appropriate weight. The DDF Loss also weights matched and unmatched predictions according to their quantity, balancing their overall contribution and individual losses. This approach results in more stable and effective distillation. The Decoupled Distillation Focal Loss $\mathcal{L}_{\text{DDF}}$ is then formulated as:

$$\mathcal{L}_{\text{DDF}} = T^2 \sum_{l=1}^{L-1} \left( \sum_{k=1}^{K_m} \alpha_k \cdot \textbf{KL}\left(\textbf{Pr}^l(n)_k, \textbf{Pr}^L(n)_k\right) + \sum_{k=1}^{K_u} \beta_k \cdot \textbf{KL}\left(\textbf{Pr}^l(n)_k, \textbf{Pr}^L(n)_k\right) \right)$$

$$\alpha_k = \text{IoU}_k \cdot \frac{\sqrt{K_m}}{\sqrt{K_m} + \sqrt{K_u}}, \quad \beta_k = \text{Conf}_k \cdot \frac{\sqrt{K_u}}{\sqrt{K_m} + \sqrt{K_u}}, \tag{6}$$

where **KL** represents the Kullback-Leibler divergence (Hinton et al., 2015), and $T$ is the temperature parameter used for smoothing logits. The distillation loss for the $k$-th matched prediction is weighted by $\alpha_k$, where $K_m$ and $K_u$ are the numbers of matched and unmatched predictions, respectively. For the $k$-th unmatched prediction, the weight is $\beta_k$, with $\text{Conf}_k$ denoting the classification confidence.

## 5 EXPERIMENTS

### 5.1 EXPERIMENT SETUP

To validate the effectiveness of our proposed methods, we conduct experiments on the COCO (Lin et al., 2014a) and Objects365 (Shao et al., 2019) datasets. We evaluate our D-FINE using the standard COCO metrics, including Average Precision (AP) averaged over IoU thresholds from 0.50 to 0.95, as well as AP at specific thresholds (AP$_{50}$ and AP$_{75}$) and AP across different object scales: small (AP$_S$), medium (AP$_M$), and large (AP$_L$). Additionally, we provide model efficiency metrics by reporting the number of parameters (#Params.), computational cost (GFLOPs), and end-to-end latency. The latency is measured using TensorRT FP16 on an NVIDIA T4 GPU.

Table 1: Performance comparison of various real-time object detectors on COCO `val2017`.

| Model | #Params. | GFLOPs | Latency (ms) | $AP^{val}$ | $AP^{val}_{50}$ | $AP^{val}_{75}$ | $AP^{val}_{S}$ | $AP^{val}_{M}$ | $AP^{val}_{L}$ |
|---|---|---|---|---|---|---|---|---|---|
| *Non-end-to-end Real-time Object Detectors* | | | | | | | | | |
| YOLOv6-L | 59M | 150 | 9.04 | 52.8 | 70.3 | 57.7 | 34.4 | 58.1 | 70.1 |
| YOLOv7-L | 36M | 104 | 16.81 | 51.2 | 69.7 | 55.5 | 35.2 | 55.9 | 66.7 |
| YOLOv7-X | 71M | 189 | 21.57 | 52.9 | 71.1 | 57.4 | 36.9 | 57.7 | 68.6 |
| YOLOv8-L | 43M | 165 | 12.31 | 52.9 | 69.8 | 57.5 | 35.3 | 58.3 | 69.8 |
| YOLOv8-X | 68M | 257 | 16.59 | 53.9 | 71.0 | 58.7 | 35.7 | 59.3 | 70.7 |
| YOLOv9-C | 25M | 102 | 10.66 | 53.0 | 70.2 | 57.8 | 36.2 | 58.5 | 69.3 |
| YOLOv9-E | 57M | 189 | 20.53 | 55.6 | 72.8 | 60.6 | 40.2 | 61.0 | 71.4 |
| Gold-YOLO-L | 75M | 152 | 9.21 | 53.3 | 70.9 | - | 33.8 | 58.9 | 69.9 |
| RTMDet-L | 52M | 80 | 14.23 | 51.3 | 68.9 | 55.9 | 33.0 | 55.9 | 68.4 |
| RTMDet-X | 95M | 142 | 21.59 | 52.8 | 70.4 | 57.2 | 35.9 | 57.3 | 69.1 |
| YOLO11-L | 25M | 87 | 10.28 | 53.4 | 70.1 | 58.2 | 35.6 | 59.1 | 69.2 |
| YOLO11-X | 57M | 195 | 14.39 | 54.7 | 71.6 | 59.5 | 37.7 | 59.7 | 70.2 |
| YOLO11-L* | 25M | 87 | 6.31 | 52.9 | 69.4 | 57.7 | 35.2 | 58.7 | 68.8 |
| YOLO11-X* | 57M | 195 | 10.52 | 54.1 | 70.8 | 58.9 | 37.0 | 59.2 | 69.7 |
| *End-to-end Real-time Object Detectors* | | | | | | | | | |
| YOLOv10-L | 24M | 120 | 7.66 | 53.2 | 70.1 | 58.1 | 35.8 | 58.5 | 69.4 |
| YOLOv10-X | 30M | 160 | 10.74 | 54.4 | 71.3 | 59.3 | 37.0 | 59.8 | 70.9 |
| RT-DETR-R50 | 42M | 136 | 9.12 | 53.1 | 71.3 | 57.7 | 34.8 | 58.0 | 70.0 |
| RT-DETR-R101 | 76M | 259 | 13.61 | 54.3 | 72.7 | 58.6 | 36.0 | 58.8 | 72.1 |
| RT-DETR-HG-L | 32M | 107 | 9.25 | 53.0 | 71.7 | 57.3 | 34.6 | 57.4 | 71.2 |
| RT-DETR-HG-X | 67M | 234 | 14.01 | 54.8 | 73.1 | 59.4 | 35.7 | 59.6 | 72.9 |
| RT-DETRv2-L | 42M | 136 | 9.15 | 53.4 | 71.6 | 57.4 | 36.1 | 57.9 | 70.8 |
| RT-DETRv2-X | 76M | 259 | 13.66 | 54.3 | 72.8 | 58.8 | 35.8 | 58.8 | 72.1 |
| RT-DETRv3-L | 42M | 136 | 9.12 | 53.4 | 71.7 | - | - | - | - |
| RT-DETRv3-X | 76M | 259 | 13.61 | 54.6 | 73.1 | - | - | - | - |
| LW-DETR-L | 47M | 72 | 8.21 | 49.5 | - | - | - | - | - |
| LW-DETR-X | 118M | 174 | 16.06 | 53.0 | - | - | - | - | - |
| **D-FINE-L** (Ours) | 31M | 91 | 8.07 | **54.0** | 71.6 | 58.4 | 36.5 | 58.0 | 71.9 |
| **D-FINE-X** (Ours) | 62M | 202 | 12.89 | **55.8** | 73.7 | 60.2 | 37.3 | 60.5 | 73.4 |
| *End-to-end Real-time Object Detectors (Pretrained on Objects365)* | | | | | | | | | |
| YOLOv10-L | 24M | 120 | 7.66 | 54.0 | 71.0 | 58.9 | 36.5 | 59.2 | 70.5 |
| YOLOv10-X | 30M | 160 | 10.74 | 54.9 | 71.9 | 59.8 | 37.6 | 60.2 | 71.7 |
| RT-DETR-R50 | 42M | 136 | 9.12 | 55.3 | 73.4 | 60.1 | 37.9 | 59.9 | 71.8 |
| RT-DETR-R101 | 76M | 259 | 13.61 | 56.2 | 74.6 | 61.3 | 38.3 | 60.5 | 73.5 |
| LW-DETR-L | 47M | 72 | 8.21 | 56.1 | 74.6 | 60.9 | 37.2 | 60.4 | 73.0 |
| LW-DETR-X | 118M | 174 | 16.06 | 58.3 | 76.9 | 63.3 | 40.9 | 63.3 | 74.8 |
| **D-FINE-L** (Ours) | 31M | 91 | 8.07 | **57.1** | 74.7 | 62.0 | 40.0 | 61.5 | 74.2 |
| **D-FINE-X** (Ours) | 62M | 202 | 12.89 | **59.3** | 76.8 | 64.6 | 42.3 | 64.2 | 76.4 |

⋆ : NMS is tuned with a confidence threshold of 0.01.

## 5.2 COMPARISON WITH REAL-TIME DETECTORS

Table 1 provides a comprehensive comparison between D-FINE and various real-time object detectors on COCO `val2017`. D-FINE achieves an excellent balance between efficiency and accuracy across multiple metrics. Specifically, **D-FINE-L** attains an AP of 54.0% with 31M parameters and 91 GFLOPs, maintaining a low latency of 8.07 ms. Additionally, **D-FINE-X** achieves an AP of 55.8% with 62M parameters and 202 GFLOPs, operating with a latency of 12.89 ms. As depicted in Figure 1, which shows scatter plots of latency vs. AP, parameter count vs. AP, and FLOPs vs. AP, D-FINE consistently outperforms other state-of-the-art models across all key dimensions. **D-FINE-L** achieves a higher AP (54.0%) compared to YOLOv10-L (53.2%), RT-DETR-R50 (53.1%), and LW-DETR-X (53.0%), while requiring fewer computational resources (91 GFLOPs vs. 120, 136, and 174). Similarly, **D-FINE-X** surpasses YOLOv10-X and RT-DETR-R101 by achieving superior performance (55.8% AP vs. 54.4% and 54.3%) and demonstrating greater efficiency in terms of lower parameter count, GFLOPs, and latency.

We further pretrain D-FINE and YOLOv10 on the Objects365 dataset (Shao et al., 2019), before finetuning them on COCO. After pretraining, both **D-FINE-L** and **D-FINE-X** exhibit significant performance improvements, achieving AP of 57.1% and 59.3%, respectively. These enhancements

Table 2: Performance comparison of various real-time object detectors on CrowdHuman.

| Model | #Params. | GFLOPs | Latency (ms) | $AP^{val}$ | $AP^{val}_{50}$ | $AP^{val}_{75}$ | $AP^{val}_{S}$ | $AP^{val}_{M}$ | $AP^{val}_{L}$ |
|---|---|---|---|---|---|---|---|---|---|
| YOLOv8-S | 11M | 29 | 6.96 | 50.4 | 78.1 | - | - | - | - |
| YOLOv9-S | 7M | 26 | 8.02 | 50.5 | 77.1 | - | - | - | - |
| YOLOv10-S | 7M | 22 | 2.65 | 50.6 | 78.6 | - | - | - | - |
| RT-DETR-R18 | 20M | 61 | 4.63 | 54.1 | 82.3 | - | - | - | - |
| RT-DETRv2-S | 20M | 60 | 4.59 | 54.2 | 82.6 | 52.2 | 19.4 | 36.8 | 47.2 |
| **D-FINE-S** (Ours) | 10M | 25 | 3.49 | **55.5** | **87.1** | 60.1 | 28.4 | 44.5 | 52.7 |

Table 3: Effectiveness of FDR and GO-LSD across various DETR models on COCO `val2017`.

| Model | #Params. | #Epochs | $AP^{val}$ | $AP^{val}_{50}$ | $AP^{val}_{75}$ | $AP^{val}_{S}$ | $AP^{val}_{M}$ | $AP^{val}_{L}$ |
|---|---|---|---|---|---|---|---|---|
| Deformable-DETR | 40M | 12 | 43.7 | 62.2 | 46.9 | 26.4 | 46.4 | 57.9 |
| + FDR & GO-LSD | 40M | 12 | 47.1 (+3.4) | 64.7 | 50.8 | 29.0 | 50.3 | 62.8 |
| DAB-DETR | 48M | 12 | 44.2 | 62.5 | 47.3 | 27.5 | 47.1 | 58.6 |
| + FDR & GO-LSD | 48M | 12 | 49.5 (+5.3) | 67.2 | 54.1 | 31.8 | 53.2 | 63.3 |
| DN-DETR | 48M | 12 | 46.0 | 64.8 | 49.9 | 27.7 | 49.1 | 62.3 |
| + FDR & GO-LSD | 48M | 12 | 49.7 (+3.7) | 67.5 | 54.4 | 31.8 | 53.4 | 63.8 |
| DINO | 47M | 12 | 49.0 | 66.6 | 53.5 | 32.0 | 52.3 | 63.0 |
| + FDR & GO-LSD | 47M | 12 | 51.6 (+2.6) | 68.6 | 56.3 | 33.8 | 55.6 | 65.3 |
| DINO | 47M | 24 | 50.4 | 68.3 | 54.8 | 33.3 | 53.7 | 64.8 |
| + FDR & GO-LSD | 47M | 24 | 52.4 (+2.0) | 69.5 | 56.9 | 34.6 | 55.7 | 66.2 |

enable them to outperform YOLOv10-L and YOLOv10-X by 3.1% and 4.4% AP, thereby positioning them as the top-performing models in this comparison. What's more, following the pretraining protocol of YOLOv8 (Glenn., 2023), YOLOv10 is pretrained on Objects365 for 300 epochs. In contrast, D-FINE requires only 21 epochs to achieve its substantial performance gains. These findings corroborate the conclusions of LW-DETR (Chen et al., 2024), demonstrating that DETR-based models benefit substantially more from pretraining compared to other detectors like YOLOs.

Table 2 compares D-FINE-S with state-of-the-art real-time detectors on the CrowdHuman dataset (Shao et al., 2018). D-FINE-S achieves the highest $AP^{val}$ (55.5%) and $AP^{val}_{50}$ (87.1%) among all models, with only 10M parameters, 25 GFLOPs, and a latency of 3.49ms.

## 5.3 EFFECTIVENESS ON VARIOUS DETR MODELS

Table 3 demonstrates the effectiveness of FDR and GO-LSD across multiple DETR-based object detectors on COCO `val2017`. These methods are designed for flexibility and can be seamlessly integrated into Deformable DETR, DAD-DETR, DN-DETR, and DINO, consistently boosting detection accuracy by 2.0% to 5.3% without increasing the number of parameters and computational burden. These results highlight the effectiveness of FDR and GO-LSD in enhancing localization precision and maximizing efficiency, demonstrating their adaptability and substantial impact across various end-to-end detection frameworks.

## 5.4 ABLATION STUDY

### 5.4.1 THE ROADMAP TO D-FINE

Table 4 showcases the stepwise progression from the baseline model (RT-DETR-HGNetv2-L (Zhao et al., 2024)) to our proposed D-FINE framework. Starting with the baseline metrics of 53.0% AP, 32M parameters, 110 GFLOPs, and 9.25 ms latency, we first remove all the decoder projection layers. This modification reduces GFLOPs to 97 and cuts the latency to 8.02 ms, although it decreases AP to 52.4%. To address this drop, we introduce the Target Gating Layer, which recovers the AP to 52.8% with only a marginal increase in computational cost.

Table 4: Step-by-step modifications from baseline model to D-FINE. Each step shows changes in AP, the number of parameters, latency, and FLOPs.

| Model | $AP^{val}$ | #Params. | Latency (ms) | GFLOPs |
|---|---|---|---|---|
| baseline: RT-DETR-HGNetv2-L (Zhao et al., 2024) | 53.0 | 32M | 9.25 | 110 |
| Remove Decoder Projection Layers | 52.4 | 32M | 8.02 | 97 |
| **+ Target Gating Layers** | 52.8 | 33M | 8.15 | 98 |
| Encoder CSP layers → GELAN (Wang & Liao, 2024) | 53.5 | 46M | 10.69 | 167 |
| Reduce Hidden Dimension in GELAN by half | 52.8 | 31M | 8.01 | 91 |
| Uneven Sampling Points (S: 3, M: 6, L: 3) | 52.9 | 31M | 7.90 | 91 |
| RT-DETRv2 Training Strategy (Lv et al., 2024) | 53.0 | 31M | 7.90 | 91 |
| **+ FDR** | 53.5 | 31M | 8.07 | 91 |
| **+ GO-LSD** | **54.0** (+1.0) | **31M** (-3%) | **8.07**(-13%) | **91** (-17%) |

Table 5: Distillation methods comparison in terms of performance, training time, and GPU memory usage. GO-LSD achieves the highest $AP^{val}$ with minimal additional training cost.

| Methods | $AP^{val}$ | Time/Epoch | Memory | FDR-based Methods | $AP^{val}$ | Time/Epoch | Memory |
|---|---|---|---|---|---|---|---|
| baseline | 53.0 | 29min | 8552M | baseline + FDR | 53.8 | 30min | 8730M |
| Logit Mimicking | 52.6 | 31min | 8554M | Localization Distill. | 53.7 | 31min | 8734M |
| Feature Imitation | 52.9 | 31min | 8554M | GO-LSD | **54.5** | 31min | 8734M |

The Target Gating Layer is strategically placed after the decoder's cross-attention module, replacing the residual connection. It allows queries to dynamically switch their focus on different targets across layers, effectively preventing information entanglement. The mechanism operates as follows:

$$\mathbf{x} = \sigma\left([\mathbf{x_1}, \mathbf{x_2}]\,\mathbf{W}^T + \mathbf{b}\right)_1 \cdot \mathbf{x_1} + \sigma\left([\mathbf{x_1}, \mathbf{x_2}]\,\mathbf{W}^T + \mathbf{b}\right)_2 \cdot \mathbf{x_2} \qquad (7)$$

where $\mathbf{x_1}$ represents the previous queries and $\mathbf{x_2}$ is the cross-attention result. $\sigma$ is the sigmoid activation function applied to the concatenated outputs, and $[.]$ represents the concatenation operation.

Next, we replace the encoder's CSP layers with GELAN layers (Wang & Liao, 2024). This substitution increases AP to 53.5% but also raises the parameter count, GFLOPs, and latency. To mitigate the increased complexity, we reduce the hidden dimension of GELAN, which balances the model's complexity and maintains AP at 52.8% while improving efficiency. We further optimize the sampling points by implementing uneven sampling across different scales (S: 3, M: 6, L: 3), which slightly increases AP to 52.9%. However, alternative sampling combinations such as (S: 6, M: 3, L: 3) and (S: 3, M: 3, L: 6) result in a minor performance drop of 0.1% AP. Adopting the RT-DETRv2 training strategy (Lv et al., 2024) (see Appendix A.1.1 for details) enhances AP to 53.0% without affecting the number of parameters or latency. Finally, the integration of FDR and GO-LSD modules elevates AP to 54.0%, achieving a 13% reduction in latency and a 17% reduction in GFLOPs compared to the baseline model. These incremental modifications demonstrate the robustness and effectiveness of our D-FINE framework.

### 5.4.2 COMPARISON OF DISTILLATION METHODS

Table 5 compares different distillation methods based on performance, training time, and GPU memory usage. The baseline model achieves an AP of 53.0%, with a training time of 29 minutes per epoch and memory usage of 8552 MB on four NVIDIA RTX 4090 GPUs. Due to the instability of one-to-one matching in DETR, traditional distillation techniques like Logit Mimicking and Feature Imitation do not improve performance; Logit Mimicking reduces AP to 52.6%, while Feature Imitation achieves 52.9%. Incorporating our FDR module increases AP to 53.8% with minimal additional training cost. Applying vanilla Localization Distillation (Zheng et al., 2022) further increases AP to 53.7%. Our GO-LSD method achieves the highest AP of 54.5%, with only a 6% increase in training time and a 2% rise in memory usage compared to the baseline. Notably, no lightweight optimizations are applied in this comparison, focusing purely on distillation performance.

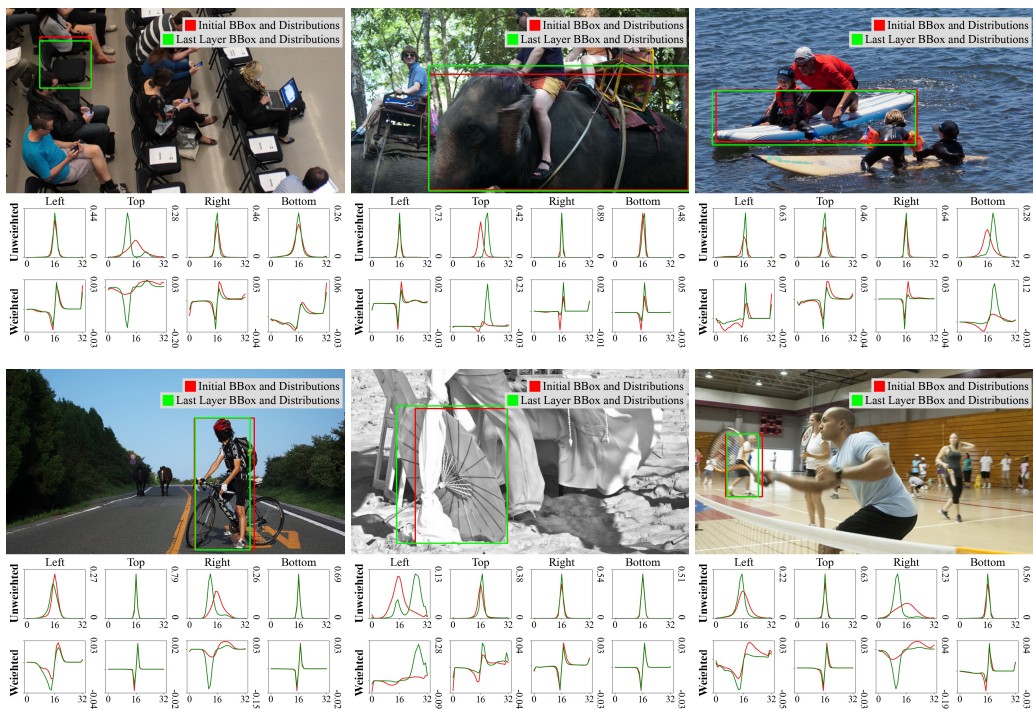

Figure 4: Visualization of FDR across detection scenarios with initial and refined bounding boxes, along with unweighted and weighted distributions, highlighting improved localization accuracy.

## 5.5 VISUALIZATION ANALYSIS

Figure 4 illustrates the process of FDR across various detection scenarios. We display the filtered detection results with two bounding boxes overlaid on the images. The red boxes represent the initial predictions from the first decoder layer, while the green boxes denote the refined predictions from the final decoder layer. The final predictions align more closely with the target objects. The first row under the images shows the unweighted probability distributions for the four edges (left, top, right, bottom). The second row shows the weighted distributions, where the weighting function $W(n)$ has been applied. The red curves represent the initial distributions, while the green curves show the final, refined distributions. The weighted distributions emphasize finer adjustments near accurate predictions and allow for enabling rapid changes for larger adjustments, further illustrating how FDR refines the offsets of initial bounding boxes, leading to increasingly precise localization.

## 6 CONCLUSION

In this paper, we introduce D-FINE, a powerful real-time object detector that redefines the bounding box regression task in DETR models through Fine-grained Distribution Refinement (FDR) and Global Optimal Localization Self-Distillation (GO-LSD). Experimental results on the COCO dataset demonstrate that D-FINE achieves state-of-the-art accuracy and efficiency, surpassing all existing real-time detectors. **Limitation and Future Work:** However, the performance gap between lighter D-FINE models and other compact models remains small. One possible reason is that shallow decoder layers may yield less accurate final-layer predictions, limiting the effectiveness of distilling localization knowledge into earlier layers. Addressing this challenge necessitates enhancing the localization capabilities of lighter models without increasing inference latency. Future research could investigate advanced architectural designs or novel training paradigms that allow for the inclusion of additional sophisticated decoder layers during training while maintaining lightweight inference by simply discarding them at test time. We hope D-FINE inspires further advancements in this area.

ACKNOWLEDGMENTS

This work was supported in part by the National Natural Science Foundation of China under Grants 62472399, 62021001, and 62032006.

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

## A APPENDIX

### A.1 IMPLEMENTATION DETAILS

#### A.1.1 HYPERPARAMETER CONFIGURATIONS

Table 6 summarizes the hyperparameter configurations for the D-FINE models. All variants use HGNetV2 backbones pretrained on ImageNet (Cui et al., 2021; Russakovsky et al., 2015) and the AdamW optimizer. D-FINE-X is set with an embedding dimension of 384 and a feedforward dimension of 2048, while the other models use 256 and 1024, respectively. The D-FINE-X and D-FINE-L have 6 decoder layers, while D-FINE-M and D-FINE-S have 4 and 3 decoder layers. The GELAN module progressively reduces hidden dimension and depth from D-FINE-X with 192 dimensions and 3 layers to D-FINE-S with 64 dimensions and 1 layer. The base learning rate and weight decay for D-FINE-X and D-FINE-L are $2.5 \times 10^{-4}$ and $1.25 \times 10^{-4}$, respectively, while D-FINE-M and D-FINE-S use $2 \times 10^{-4}$ and $1 \times 10^{-4}$. Smaller models also have higher backbone learning rates than larger models. The total batch size is 32 across all variants. Training schedules include 72 epochs with advanced augmentation (`RandomPhotometricDistort`, `RandomZoomOut`, `RandomIoUCrop`, and `RMultiScaleInput`) followed by 2 epochs without advanced augmentation for D-FINE-X and D-FINE-L, and 120 epochs with advanced augmentation followed by 4 epochs without advanced augmentation for D-FINE-M and D-FINE-S (RT-DETRv2 Training Strategy (Lv et al., 2024) in Table 4). The number of pretraining epochs is 21 for D-FINE-X and D-FINE-L models, while for D-FINE-M and D-FINE-S models, it ranges from 28 to 29 epochs.

Table 6: Hyperparameter configurations for different D-FINE models.

| Setting | D-FINE-X | D-FINE-L | D-FINE-M | D-FINE-S |
|---|---|---|---|---|
| Backbone Name | HGNetv2-B5 | HGNetv2-B4 | HGNetv2-B2 | HGNetv2-B0 |
| Optimizer | AdamW | AdamW | AdamW | AdamW |
| Embedding Dimension | 384 | 256 | 256 | 256 |
| Feedforward Dimension | 2048 | 1024 | 1024 | 1024 |
| GELAN Hidden Dimension | 192 | 128 | 128 | 64 |
| GELAN Depth | 3 | 3 | 2 | 1 |
| Decoder Layers | 6 | 6 | 4 | 3 |
| Queries | 300 | 300 | 300 | 300 |
| $a, c$ in $W(n)$ | 0.5, 0.125 | 0.5, 0.25 | 0.5, 0.25 | 0.5, 0.25 |
| Bin Number $N$ | 32 | 32 | 32 | 32 |
| Sampling Point Number | (S: 3, M: 6, L: 3) | (S: 3, M: 6, L: 3) | (S: 3, M: 6, L: 3) | (S: 3, M: 6, L: 3) |
| Temperature $T$ | 5 | 5 | 5 | 5 |
| Base LR | 2.5e-4 | 2.5e-4 | 2e-4 | 2e-4 |
| Backbone LR | 2.5e-6 | 1.25e-5 | 2e-5 | 1e-4 |
| Weight Decay | 1.25e-4 | 1.25e-4 | 1e-4 | 1e-4 |
| Weight of $\mathcal{L}_{\text{VFL}}$ | 1 | 1 | 1 | 1 |
| Weight of $\mathcal{L}_{\text{BBox}}$ | 5 | 5 | 5 | 5 |
| Weight of $\mathcal{L}_{\text{GIOU}}$ | 2 | 2 | 2 | 2 |
| Weight of $\mathcal{L}_{\text{FGL}}$ | 0.15 | 0.15 | 0.15 | 0.15 |
| Weight of $\mathcal{L}_{\text{DDF}}$ | 1.5 | 1.5 | 1.5 | 1.5 |
| Total Batch Size | 32 | 32 | 32 | 32 |
| EMA Decay | 0.9999 | 0.9999 | 0.9999 | 0.9999 |
| Epochs (w/ + w/o Adv. Aug.) | 72 + 2 | 72 + 2 | 120 + 4 | 120 + 4 |
| Epochs (Pretrain + Finetune) | 21 + 31 | 21 + 32 | 29 + 49 | 28 + 58 |

#### A.1.2 DATASETS SETTINGS

For pretraining, following the approach in (Chen et al., 2022b; Zhang et al., 2022; Chen et al., 2024), we combine the images from the Objects365 (Shao et al., 2019) train set with the validate set, excluding the first 5k images. To further improve training efficiency, we resize all images with resolutions exceeding $640 \times 640$ down to $640 \times 640$ beforehand. We use the standard COCO2017 (Lin et al., 2014b) data splitting policy, training on COCO `train2017`, and evaluating on COCO `val2017`.

## A.2 VISUALIZATION OF D-FINE PREDICTIONS

Figure 5 demonstrates the robustness of the D-FINE-X model, visualizing its predictions in various challenging scenarios. These include occlusion, low-light conditions, motion blur, depth of field effects, rotation, and densely populated scenes with numerous objects in close proximity. Despite these difficulties, the model accurately identifies and localizes objects, such as animals, vehicles, and people. This visualization highlights the model's ability to handle complex real-world conditions while maintaining robust detection performance.

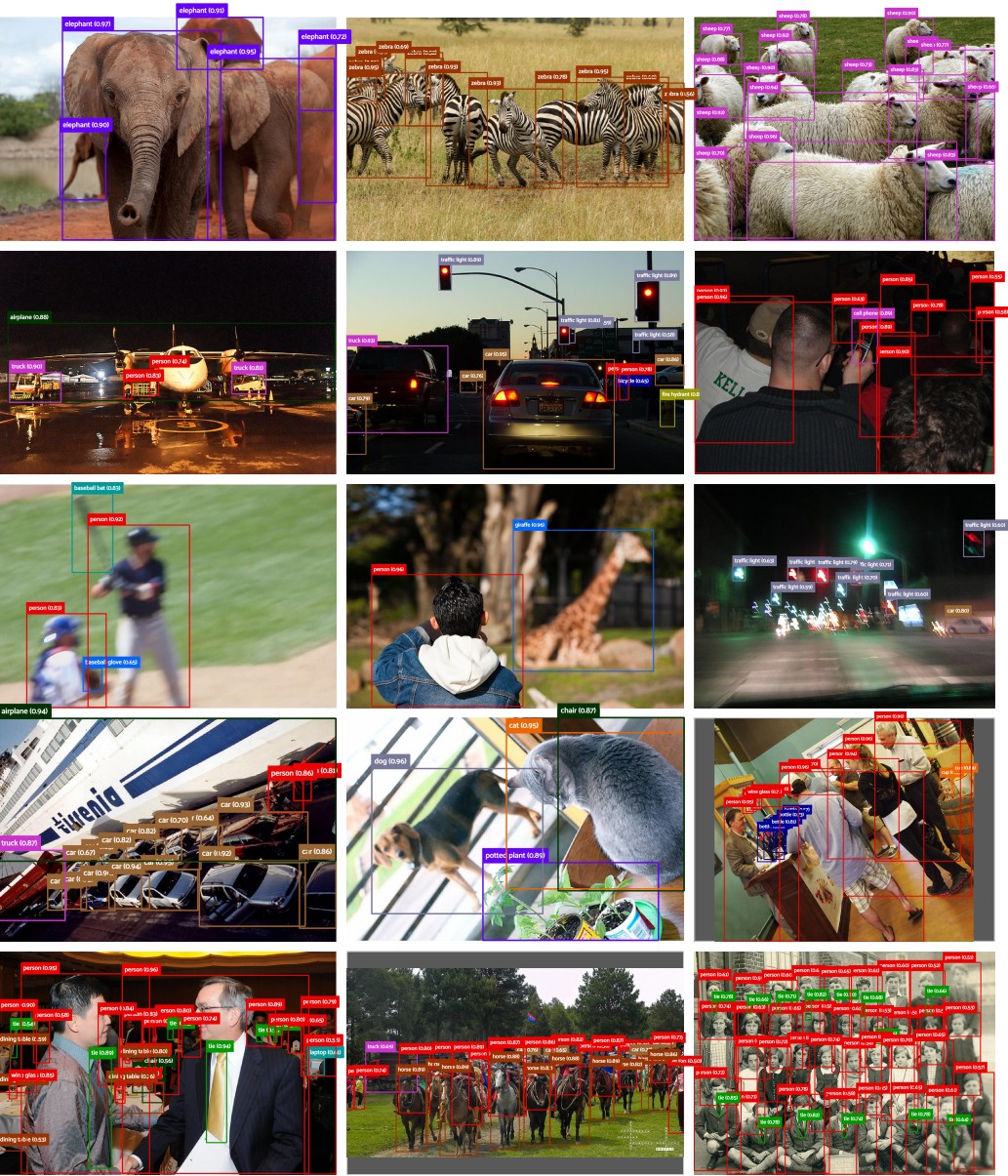

Figure 5: Visualization of D-FINE-X (without pre-training on Objects365) predictions under challenging conditions, including occlusion, low light, motion blur, depth of field effects, rotation, and densely populated scenes (confidence threshold=0.5).

### A.3 Comparison with Lighter Detectors

Table 7 presents a comprehensive comparison of D-FINE models with various lightweight real-time object detectors in the S and M model sizes on the COCO `val2017`. D-FINE-S achieves an impressive AP of 48.5%, surpassing other lightweight models such as Gold-YOLO-S (46.4%) and RT-DETRv2-S (48.1%), while maintaining a low latency of 3.49 ms with only 10.2M parameters and 25.2 GFLOPs. Pretraining on Objects365 further boosts D-FINE-S to 50.7%, marking an improvement of +2.2%. Similarly, D-FINE-M attains an AP of 52.3% with 19.2M parameters and 56.6 GFLOPs at 5.62 ms, outperforming YOLOv10-M (51.1%) and RT-DETRv2-M (49.9%). Pretraining on Objects365 consistently enhances D-FINE-M, yielding a +2.8% gain. These results demonstrate that D-FINE models strike an excellent balance between accuracy and efficiency, consistently surpassing other state-of-the-art lightweight detectors while preserving real-time performance.

Table 7: Performance comparison of S and M sized real-time object detectors on COCO `val2017`.

| Model | #Params. | GFLOPs | Latency (ms) | $AP^{val}$ | $AP^{val}_{50}$ | $AP^{val}_{75}$ | $AP^{val}_{S}$ | $AP^{val}_{M}$ | $AP^{val}_{L}$ |
|---|---|---|---|---|---|---|---|---|---|
| *Non-end-to-end Real-time Object Detectors* | | | | | | | | | |
| YOLOv6-S | 7M | 17 | 3.62 | 45.0 | 61.8 | 48.9 | 24.3 | 50.2 | 62.7 |
| YOLOv6-M | 35M | 86 | 5.48 | 50.0 | 66.9 | 54.6 | 30.6 | 55.4 | 67.3 |
| YOLOv8-S | 11M | 29 | 6.96 | 44.9 | 61.8 | 48.6 | 25.7 | 49.9 | 61.0 |
| YOLOv8-M | 26M | 79 | 9.66 | 50.2 | 67.2 | 54.6 | 32.0 | 55.7 | 66.4 |
| YOLOv9-S | 7M | 26 | 8.02 | 44.9 | 61.8 | 48.6 | 25.7 | 49.9 | 61.0 |
| YOLOv9-M | 20M | 76 | 10.15 | 50.2 | 67.2 | 54.6 | 32.0 | 55.7 | 66.4 |
| Gold-YOLO-S | 22M | 46 | 2.01 | 46.4 | 63.4 | - | 25.3 | 51.3 | 63.6 |
| Gold-YOLO-M | 41M | 88 | 3.21 | 51.1 | 68.5 | - | 32.3 | 56.1 | 68.6 |
| RTMDet-S | 9M | 15 | 7.77 | 44.6 | 61.9 | 48.1 | 24.9 | 48.5 | 62.5 |
| RTMDet-M | 25M | 39 | 10.62 | 49.4 | 66.8 | 53.7 | 30.3 | 53.9 | 66.2 |
| YOLO11-S | 9M | 22 | 6.81 | 46.6 | 63.4 | 50.3 | 28.7 | 51.3 | 64.1 |
| YOLO11-M | 20M | 68 | 8.79 | 51.2 | 67.9 | 55.3 | 33.0 | 56.7 | 67.5 |
| YOLO11-S⋆ | 9M | 22 | 2.86 | 47.0 | 63.9 | 50.7 | 29.0 | 51.7 | 64.4 |
| YOLO11-M⋆ | 20M | 68 | 4.95 | 51.5 | 68.5 | 55.7 | 33.4 | 57.1 | 67.9 |
| *End-to-end Real-time Object Detectors* | | | | | | | | | |
| YOLOv10-S | 7M | 22 | 2.65 | 46.3 | 63.0 | 50.4 | 26.8 | 51.0 | 63.8 |
| YOLOv10-M | 15M | 59 | 4.97 | 51.1 | 68.1 | 55.8 | 33.8 | 56.5 | 67.0 |
| RT-DETR-R18 | 20M | 61 | 4.63 | 46.5 | 63.8 | 50.4 | 28.4 | 49.8 | 63.0 |
| RT-DETR-R34 | 31M | 93 | 6.43 | 48.9 | 66.8 | 52.9 | 30.6 | 52.4 | 66.3 |
| RT-DETRv2-S | 20M | 60 | 4.59 | 48.1 | 65.1 | 57.4 | 36.1 | 57.9 | 70.8 |
| RT-DETRv2-M | 31M | 92 | 6.40 | 49.9 | 67.5 | 58.6 | 35.8 | 58.6 | 72.1 |
| RT-DETRv3-R18 | 20M | 61 | 4.63 | 48.1 | 66.2 | - | - | - | - |
| RT-DETRv3-R34 | 31M | 93 | 6.43 | 49.9 | 67.7 | - | - | - | - |
| LW-DETR-S | 15M | 17 | 3.02 | 43.6 | - | - | - | - | - |
| LW-DETR-M | 28M | 43 | 5.23 | 47.2 | - | - | - | - | - |
| **D-FINE-N** (Ours) | 4M | 7.2 | 2.12 | **42.8** | 60.2 | 45.4 | 22.8 | 46.8 | 61.9 |
| **D-FINE-S** (Ours) | 10M | 25 | 3.49 | **48.5** | 65.6 | 52.6 | 29.1 | 52.2 | 65.4 |
| **D-FINE-M** (Ours) | 19M | 57 | 5.55 | **52.3** | 69.8 | 56.4 | 33.2 | 56.5 | 70.2 |
| *End-to-end Real-time Object Detectors (Pretrained on Objects365)* | | | | | | | | | |
| RT-DETR-R18 | 20M | 61 | 4.63 | 49.2 | 66.6 | 53.5 | 33.2 | 52.3 | 64.8 |
| LW-DETR-S | 15M | 17 | 3.02 | 48.0 | 66.9 | 51.7 | 26.8 | 52.5 | 65.5 |
| LW-DETR-M | 28M | 43 | 5.23 | 52.6 | 72.0 | 56.7 | 32.6 | 57.7 | 70.7 |
| **D-FINE-S** (Ours) | 10M | 25 | 3.49 | **50.7** | 67.6 | 55.1 | 32.7 | 54.6 | 66.5 |
| **D-FINE-M** (Ours) | 19M | 57 | 5.62 | **55.1** | 72.6 | 59.7 | 37.9 | 59.4 | 71.7 |

⋆ : NMS is tuned with a confidence threshold of 0.01.

Table 8 presents a concise performance comparison on Objects365. Our D-FINE models consistently deliver high AP scores while maintaining low latency and minimal computational overhead. In particular, D-FINE-X achieves an impressive 49.5% AP with 62M parameters at 12.89 ms, and even the lightweight D-FINE-S registers a competitive 31.0% AP with only 10M parameters. These results underscore the excellent balance between accuracy and efficiency offered by the D-FINE series on the complex, large-scale dataset, outperforming both RT-DETR and YOLOv10 models.

Table 8: Performance comparison of real-time object detectors on Objects365.

| Model | #Params. | GFLOPs | Latency (ms) | $AP^{val}$ | $AP^{val}_{50}$ |
|---|---|---|---|---|---|
| RT-DETR-R18 | 20M | 61 | 4.63 | 22.9 | 31.2 |
| RT-DETR-R50 | 42M | 136 | 9.12 | 35.1 | 46.2 |
| RT-DETR-R101 | 76M | 259 | 13.61 | 36.8 | 48.3 |
| YOLOv10-L | 24M | 120 | 7.66 | 34.2 | 44.9 |
| YOLOv10-X | 30M | 160 | 10.74 | 34.9 | 45.8 |
| **D-FINE-S** (Ours) | 10M | 25 | 3.49 | **31.0** | 41.0 |
| **D-FINE-M** (Ours) | 19M | 57 | 5.55 | **38.6** | 49.7 |
| **D-FINE-L** (Ours) | 31M | 91 | 8.07 | **44.7** | 56.9 |
| **D-FINE-X** (Ours) | 62M | 202 | 12.89 | **49.5** | 62.4 |

Table 9: Hyperparameter ablation studies on D-FINE-L. $\epsilon$ is a very small value. $\widetilde{a}, \widetilde{c}$ indicate that $a$ and $c$ are learnable parameters.

| $a, c$ | $\frac{1}{4}, \frac{1}{4}$ | $\frac{1}{2}, \frac{1}{\epsilon}$ | $\mathbf{\frac{1}{2}, \frac{1}{4}}$ | $\frac{1}{2}, \frac{1}{8}$ | $1, \frac{1}{4}$ | $\widetilde{a}, \widetilde{c}$ |
|---|---|---|---|---|---|---|
| $AP^{val}$ | 52.7 | 53.0 | **53.3** | 53.2 | 53.2 | 53.1 |

| $N$ | 4 | 8 | 16 | **32** | 64 | 128 |
|---|---|---|---|---|---|---|
| $AP^{val}$ | 53.3 | 53.4 | 53.5 | **53.7** | 53.6 | 53.6 |

| $T$ | 1 | 2.5 | **5** | 7.5 | 10 | 20 |
|---|---|---|---|---|---|---|
| $AP^{val}$ | 53.2 | 53.7 | **54.0** | 53.8 | 53.7 | 53.5 |

### A.4 HYPERPARAMETER SENSITIVITY ANALYSIS

Appendix A.3 presents a subset of hyperparameter ablation studies evaluating the sensitivity of our model to key parameters in the FDR and GO-LSD modules. We examine the weighting function parameters $a$ and $c$, the number of distribution bins $N$, and the temperature $T$ used for smoothing logits in the KL divergence.

**(1)** Setting $a = \frac{1}{2}$ and $c = \frac{1}{4}$ yields the highest AP of 53.3%. Notably, treating $a$ and $c$ as learnable parameters ($\tilde{a}, \tilde{c}$) slightly decreases AP to 53.1%, suggesting that fixed values simplify the optimization process. When $c$ is extremely large, the weighting function approximates the linear function with equal intervals, resulting in a suboptimal AP of 53.0%. Additionally, values of $a$ that are too large or too small can reduce fineness or limit flexibility, adversely affecting localization precision. **(2)** Increasing the number of distribution bins improves performance, with a maximum AP of 53.7% achieved at $N = 32$. Beyond $N = 32$, no significant gain is observed. **(3)** The temperature $T$ controls the smoothing of logits during distillation. An optimal AP of 54.0% is achieved at $T = 5$, indicating a balance between softening the distribution and preserving effective knowledge transfer.

### A.5 CLARIFICATION ON THE INITIAL LAYER REFINEMENT

In the main text, we define the refined distributions at layer $l$ as:

$$\mathbf{Pr}^l(n) = \text{Softmax}\left(\Delta\text{logits}^l(n) + \text{logits}^{l-1}(n)\right), \tag{8}$$

where $\Delta\text{logits}^l(n)$ are the residual logits predicted by layer $l$, and $\text{logits}^{l-1}(n)$ are the logits from the previous layer.

For the initial layer ($l = 1$), there is no previous layer, so the formula simplifies to:

$$\mathbf{Pr}^1(n) = \text{Softmax}\left(\text{logits}^1(n)\right). \tag{9}$$

Here, $\text{logits}^1(n)$ are the logits predicted by the first layer.

This clarification ensures the formulation is consistent and mathematically rigorous for all layers.

