# OpenReview forum: "D-FINE: Redefine Regression Task of DETRs as Fine-grained Distribution Refinement"
_ICLR.cc/2025/Conference — ICLR 2025 Spotlight_

### Official Review · Reviewer_759J · 2024-10-30

**Soundness:** 4
**Presentation:** 3
**Contribution:** 4
**Rating:** 8
**Confidence:** 4

**Summary:**

D-FINE is a real-time object detection model that refines bounding box regression in Detection Transformers using Fine-grained Distribution Refinement (FDR) and Global Optimal Localization Self-Distillation (GO-LSD). These techniques enhance localization accuracy and efficiency by iteratively refining bounding boxes and transferring refined knowledge across layers. D-FINE outperforms other real-time detectors on the COCO dataset, achieving a better balance of speed and precision with minimal added costs.

**Strengths:**

1.	D-FINE surpasses all existing real-time detectors with negligible extra parameters and training costs. Source code is provided. This method appears to be highly solid and reproducible.
2.	Table 2 illuminates that the proposed FDR and GO-LSD are efficient for a series of DETR models, demonstrating the robustness of the module.
3.	The proposed method is both innovative and systematic.

**Weaknesses:**

1.	In Table 3, FDR and GO-LSD modules have a 1% mAP improvement in total for D-FINE, which is noticeably less than the improvement shown in Table 2.
2.	The language in the article needs refinement, as some phrases may lead to ambiguity. For example, line 206-208, “Initially, the first …” and line 209, “…one for each edge”.
3.	Variable names in the Method need to be consistent. For example, line 213, {W, H} and line 139 {w,h}. The lowercase ‘w’ is preferable, as the uppercase ‘W’ repeats the notation used for the weighting function below.

**Questions:**

1.	In Table 3, FDR and GO-LSD modules have a 1% mAP improvement in total for D-FINE. However, according to table 2, these two modules improve accuracy by 2% at least. Can the author explain this phenomenon?

---

> ### Author Response · Authors · 2024-11-19
>
> Thank you for your detailed review and for highlighting the strengths of our work. We are pleased that you find our method innovative, systematic, and highly reproducible.
>
> **Regarding the smaller mAP improvement in Table 3 compared to Table 2:**
>
> Thank you for your observation. It's worth noting that as the baseline mAP increases, achieving further improvements becomes more challenging—a phenomenon often referred to as diminishing returns. Additionally, the models in Table 2 (now updated as Table 3) have not undergone the lightweight optimizations applied to the models in Table 3 (now updated as Table 4). When these lightweight modifications are excluded, as shown in Table 5, our FDR and GO-LSD modules improve the baseline model (RT-DETR-HGNetv2-L) from 53.0% to 54.5% mAP, demonstrating their effectiveness in enhancing performance.
>
> **Regarding the need for language refinement due to potential ambiguity:**
>
> Thank you for pointing out the ambiguities in our phrasing, particularly in lines 206–208 ("Initially, the first ...") and line 209 ("... one for each edge"). We appreciate your feedback and have revised the text for clarity to ensure that our explanations are precise and easy to understand. We will continue to review the manuscript to address any similar issues.
>
> **Regarding consistency in variable names in the Method section:**
>
> Thank you for your valuable suggestion. Initially, we intended to differentiate the notations for width and height between the **Preliminaries** and **Method** sections. However, we agree that consistency is important to avoid confusion. We have now unified all such notations to lowercase letters (i.e., {w, h}) throughout the manuscript for consistency and clarity.
>
> ---
>
> Thank you again for your constructive feedback. We believe that these revisions enhance the clarity and quality of our paper, and we appreciate your assistance in improving it.

---

### Official Review · Reviewer_5Uux · 2024-11-01

**Soundness:** 3
**Presentation:** 3
**Contribution:** 3
**Rating:** 8
**Confidence:** 4

**Summary:**

The paper introduces D-FINE, a real-time object detection model that enhances DETR by redefining bounding box regression with Fine-grained Distribution Refinement (FDR) and Global Optimal Localization Self-Distillation (GO-LSD). The FDR transforms bounding box predictions into refined probability distributions, while GO-LSD uses refined localization knowledge to improve earlier layers' predictions. Experiments on the COCO dataset show that D-FINE achieves good performance and high speed, positioning it as a strong competitor to existing real-time detectors.

**Strengths:**

1. The paper presents D-FINE, an enhancement to the DETR framework that tackles bounding box regression with a novel fine-grained distribution refinement and self-distillation mechanism.

2. The experimental results on the COCO dataset demonstrate competitive performance, showing that D-FINE outperforms many existing real-time object detectors, with a favorable trade-off between speed and accuracy. Additionally, the paper’s ablation studies and hyperparameter tuning offer insightful explanations of the model’s design choices.

3. The experiments and comparisons with various DETR-based models are thorough. The inclusion of both small and large models, alongside real-world visualizations, strengthens the validity of the claims about D-FINE's performance across diverse conditions.

**Weaknesses:**

The experiments focus primarily on the COCO dataset. Evaluating D-FINE across a broader range of datasets could strengthen its generalizability claims and confirm its robustness across different object detection contexts, such as crowded scene CrowdHuman and long tail scene LVIS.

**Questions:**

See weaknesses

---

> ### Author Response · Authors · 2024-11-19
>
> Thank you for your valuable suggestion. We agree that evaluating D-FINE across a broader range of datasets would strengthen our claims about generalizability and robustness. In response, we have updated the manuscript to include comparisons of D-FINE with other real-time object detectors on both the CrowdHuman and Objects365 datasets (see Table 2 and Table 8):
>
> ### Table 2: Performance Comparison of Real-Time Object Detectors on CrowdHuman
>
> | Model            | #Params. | GFLOPs | Latency (ms) | AP$^{val}$ | AP$^{val}_{50}$ |
> |-------------------|----------|--------|--------------|------------|-----------------|
> | YOLOv8-S         | 11M      | 29     | 6.96         | 50.4       | 78.1            |
> | YOLOv8-M         | 26M      | 79     | 9.66         | 53.9       | 80.7            |
> | YOLOv9-S         | 7M       | 26     | 8.02         | 50.5       | 77.1            |
> | YOLOv10-S        | 7M       | 22     | 2.65         | 50.6       | 78.6            |
> | RT-DETR-R18      | 20M      | 61     | 4.63         | 54.1       | 82.3            |
> | RT-DETRv2-S      | 20M      | 60     | 4.59         | 54.2       | 82.6            |
> | **D-FINE-S (Ours)** | 10M      | 25     | 3.49         | **55.5**   | **87.1**        |
>
> ### Table 8: Performance Comparison of Real-Time Object Detectors on Objects365
>
> | Model            | #Params. | GFLOPs | Latency (ms) | AP$^{val}$ | AP$^{val}_{50}$ |
> |-------------------|----------|--------|--------------|------------|-----------------|
> | RT-DETR-R18      | 20M      | 61     | 4.63         | 22.9       | 31.2            |
> | RT-DETR-R50      | 42M      | 136    | 9.12         | 35.1       | 46.2            |
> | RT-DETR-R101     | 76M      | 259    | 13.61        | 36.8       | 48.3            |
> | YOLOv10-L        | 24M      | 120    | 7.66         | 34.2       | 44.9            |
> | YOLOv10-X        | 30M      | 160    | 10.74        | 34.9       | 45.8            |
> | **D-FINE-S (Ours)** | 10M      | 25     | 3.49         | **31.0**   | **41.0**        |
> | **D-FINE-M (Ours)** | 19M      | 57     | 5.55         | **38.6**   | **49.7**        |
> | **D-FINE-L (Ours)** | 31M      | 91     | 8.07         | **44.7**   | **56.9**        |
> | **D-FINE-X (Ours)** | 62M      | 202    | 12.89        | **49.5**   | **62.4**        |
>
> These results demonstrate the significant superiority of D-FINE on dense scenes and large-scale complex datasets. We have also incorporated comparisons with RT-DETRv3 in Table 1 and Table 7 of the revised manuscript. We plan to further expand our evaluation with more datasets and will provide additional configurations and pretrained weights in future updates.
>
> Thank you again for your constructive feedback. We believe these enhancements improve the presentation and strengthen the generalizability claims of our work.

---

> ### Comment · Area_Chair_Q53A · 2024-11-25
> **review the rebuttal**
>
> Dear Reviewer 5Uux,
>
> Could you kindly review the rebuttal thoroughly and let us know whether the authors have adequately addressed the issues raised or if you have any further questions.
>
> Best,
>
> AC of Submission1611

---

> > ### Comment · Reviewer_5Uux · 2024-11-25
> >
> > Thanks for your reminder. I appreciate the contributions of this work, and the authors address my concerns in the rebuttal. So I would increase my score to 7.

---

> > > ### Author Response · Authors · 2024-11-25
> > >
> > > We’re delighted to hear your positive feedback! Please feel free to update your score through the 'Edit Official Review' option. Thank you!

---

### Official Review · Reviewer_hveE · 2024-11-02

**Soundness:** 3
**Presentation:** 3
**Contribution:** 3
**Rating:** 8
**Confidence:** 4

**Summary:**

The paper proposes D-FINE, which consists of two methods, Fine-grained Distribution Refinement (FDR) and Global Optimal Localization Self-Distillation (GO-LSD) to improve the performance of real-time object detectors. Based on the probability distribution representation of bbox, the paper uses multiple D-FINE head to predict the prob. distributions of bboxes on $L$ layers in one decoder. FDR uses a hand-crafted weight function to weight the prob. distribution of bbox and then iteratively refine the bboxes. GO-LSD is inspired from localization distillation, which distills the localization knowledge from the prob. distribution of bboxes of the last layer to the ones of the shallow layers. The paper also designs two losses to work with the proposed FDR and GO-LSD. Experiments on COCO benchmark show the effectiveness of the proposed method.

**Strengths:**

1. The proposed method is technically sound, which is supported by sufficient experiments.

2. The paper shows the roadmap from the baseline model to the proposed D-FINE framework, making the technical contribution clear and transparent.

3. The proposed D-FINE achieves state-of-the-art performance in real-time object detection.

**Weaknesses:**

1. It is confusing to me that in Fig. 2, it seems that D-FINE with FDR is applied to the 1-st decoder layer of the object detector. Then, within 1 decoder layer, multiple D-FINE head is used to generate the prob. distribution of bboxes. However, on lines 207-208, you mention "the first decoder layer predicts preliminary bounding boxes and preliminary probability distributions through a traditional bounding box regression head and **a D-FINE head**". In Sec. 4.2,  GO-LSD utilizes the final layer’s refined distribution predictions to distill localization knowledge into the earlier layers. One can see in Fig. 3 that the self-distillation is conducted between different decoders. Thus, is that the meaning of the word "layer" different in FDR and GO-LSD? What is the exact meaning of "layer" in the context of FDR versus GO-LSD?

2. In Fig. 3, the matched prediction and unmatched prediction are colored by green squares and yellow squares. What does the gray squares stand for? I suggest that the authors could explicitly state what the gray squares represent in the caption or legend of Figure 3.

**Questions:**

1. How many D-FINE heads are used in the 1-st decoder, e.g., the value of $L$? Is FDR applied within each decoder layer or across decoder layers?

2. If FDR is applied differently across different decoders, whether the number of D-FINE heads is consistent across all decoder layers or if it varies? How and why it differs? I suggest that the authors add this information to provide a clearer picture of the model's architecture and operation would enhance the reproducibility of the proposed method.

---

> ### Author Response · Authors · 2024-11-19
>
> Thank you for your insightful comments and for pointing out areas that needed clarification. We appreciate your thorough review and are glad that you find the proposed method technically sound and well-presented.
>
> **Regarding the potential confusion in Figure 2 and the meaning of "layer" in FDR versus GO-LSD:**
>
> We apologize for the confusion caused by Figure 2. To address this, we have redrawn Figure 2 and explicitly numbered the different decoder layers for better clarity.
>
> 1. **Clarification on Figure 2 and Decoder Layers:**
>    - In our model, every decoder layer outputs probability distributions through one D-FINE head. The first decoder layer includes an additional traditional bounding box regression head, which generates preliminary bounding boxes.
>    - This means that within each decoder layer, there is one D-FINE head, and in the first decoder layer, there is an extra traditional head. The D-FINE heads iteratively refine the probability distributions across the decoder layers.
>
> 2. **Consistency of the Term "Layer":**
>    - The term "layer" is used consistently across both Fine-grained Distribution Refinement (FDR) and Global Optimal Localization Self-Distillation (GO-LSD).
>    - Specifically, **FDR** operates across all decoder layers, refining the probability distributions iteratively within each layer.
>    - The outputs from these layers are then used in **GO-LSD** to distill localization knowledge from deeper layers to shallower ones. Thus, both FDR and GO-LSD involve operations across the same decoder layers, and the term "layer" refers to the layers of the decoder in both contexts.
>
> **Regarding Figure 3 and the representation of predictions:**
>
> Thank you for bringing this to our attention. We have revised Figure 3 to remove the potentially misleading gray squares and have unified the representation using unmatched predictions. Additionally, we have included annotations indicating the respective numbers of matched and unmatched predictions for clarity. This should make the figure more understandable and accurately convey the intended information.
>
> **Questions:**
>
> 1. **How many D-FINE heads are used in the first decoder layer, i.e., the value of \( L \)? Is FDR applied within each decoder layer or across decoder layers?**
>
>    - In our implementation, each decoder layer has one D-FINE head, so the value of \( L \) corresponds to the number of decoder layers, which is three in our case.
>    - **FDR is applied across decoder layers**, where each D-FINE head refines the probability distributions iteratively based on the previous layer's output.
>
> 2. **If FDR is applied differently across different decoders, is the number of D-FINE heads consistent across all decoder layers or does it vary? How and why does it differ?**
>
>    - The number of D-FINE heads is consistent across all decoder layers; each layer has one D-FINE head.
>    - The exception is the first decoder layer, which also includes an additional traditional regression head for generating preliminary bounding boxes.
>    - We have added this information to the manuscript to provide a clearer picture of the model's architecture and operation. This enhancement should improve the reproducibility of our proposed method.
>
> ---
>
> Thank you again for your valuable feedback. We believe that these clarifications and revisions will improve the clarity of our paper, and we appreciate your assistance in enhancing its quality.

---

> > ### Comment · Reviewer_hveE · 2024-11-25
> >
> > The authors' rebuttal address my concerns. I will keep my rating score.

---

### Official Review · Reviewer_oAYL · 2024-11-02

**Soundness:** 2
**Presentation:** 3
**Contribution:** 2
**Rating:** 6
**Confidence:** 4

**Summary:**

This paper introduces a real-time object detector based on DETR through two main components: fine-grained distribution refinement, which refines probability distributions iteratively for improved accuracy, and global optimal localization self-distillation, which optimizes localization through a bidirectional self-distillation approach.

**Strengths:**

The paper is easy to understand and well-presented. A set of experiments are conducted on the COCO and Objects365 datasets.

**Weaknesses:**

1. I do not fully agree with the claim that using fixed coordinates results in poor performance due to inadequate modelling of localization uncertainty. While the statement shows potential limitations of fixed-coordinate regression, many SOTA object detectors (Faster R-CNN and RetinaNet), represent significant advances in both speed and detection. Therefore, to say that all methods using fixed coordinates were fundamentally limited overlooks these advancements. The authors may clarify their claim by acknowledging the success of fixed-coordinate methods while explaining how the proposed approach can address these limitations

2. It has been stated, " As early predictions improve, the subsequent layers can focus on refining smaller residuals.", what do you mean by  "refining smaller residuals"?

3. In page 5, Eq. (3).  The normalizing updated logits can result in very small gradients when logits are large, making it harder for the model to learn effectively in deeper layers. This can reduce the model’s precision in refining bounding box edges. Moreover, softmax forces the output to sum to one, which can limit its ability to model complex relationships between bins and reduce its performance for handling localization uncertainty. Have you tried any other normalization methods such as Entmax normalization which well fit into DERT methods? and why softmax has been chosen over other options?

4. In Figure 4 what those sub-plots (right, left, ...) mean and represent? There is no discussion about these plots. It is not clear why these plots are shown and what they are representing. Moreover, it is unclear, what the pick value means and what different lines (red, green) are supposed to be for the best performance. The authors need to provide a more detailed caption or explanation for this figure, specifically describing what each subplot represents and the meaning of the different colored lines.

**Questions:**

Please see the weaknesses.

---

> ### Author Response · Authors · 2024-11-19
>
> Thank you for your detailed review. We appreciate your thoughtful comments.
>
> **Regarding the claim that using fixed coordinates results in poor performance due to inadequate modeling of localization uncertainty:**
>
> We absolutely acknowledge the great success of fixed-coordinate methods like Faster R-CNN and RetinaNet, which have significantly advanced both speed and detection performance. We have revised the manuscript to clarify that our approach aims to optimize by better modeling localization uncertainty, rather than suggesting that fixed-coordinate regression methods are fundamentally limited. Our method builds upon these advancements by introducing a different perspective on handling localization, which we believe complements existing approaches.
>
> **Regarding the phrase "refining smaller residuals":**
>
> As shown in Figure 2, D-FINE performs layer-wise optimization of the bounding boxes, progressively bringing them closer to the objects. By "refining smaller residuals," we mean that as the earlier layers produce more accurate bounding box predictions, the difference (or residual) between the predicted boxes and the ground truth becomes smaller. This allows the subsequent layers to focus on correcting these smaller errors rather than making large corrections, effectively fine-tuning the predictions.
>
> **Regarding the use of Softmax over other normalization methods like Entmax in Eq. (3):**
>
> 1. **Consistency with KL Divergence:** We chose to use Softmax to maintain consistency with the KL divergence calculation in Eq. (6), where Softmax and Log-Softmax are typically used to normalize logits, preserving more "knowledge" in the process.
>
> 2. **Experiments with Alternatives:** During the development of D-FINE, we experimented with several alternatives, including Sigmoid and Entmax. While Entmax also ensures the output sums to 1 (with the default parameter $\(1 < \alpha < 2\)$, where $\(\alpha = 1\)$ is equivalent to Softmax), it introduces a sparsity mechanism. The larger $\(\alpha\)$ is, the more likely small probabilities are set to zero, and larger probabilities become more pronounced. While this can simplify training and improve generalization in some cases, we preferred Softmax for D-FINE because it allows us to preserve even small probabilities. Since D-FINE involves a weighted sum of probabilities, preserving small probabilities ensures they contribute to the final bounding box prediction rather than being ignored.
>
> 3. **Computational Efficiency:** Additionally, Softmax has been widely used in numerous models, and its underlying implementation has been highly optimized. In our tests on a NVIDIA RTX 4090 GPU, for a sequence of length 10,000, the time for 1,000 iterations of Entmax was 3.0389 seconds, while for Softmax it was only 0.00472 seconds. Given that D-FINE is a real-time detector and Softmax performs well in our context, we chose Softmax for its efficiency and effectiveness.
>
> **Regarding the explanation of Figure 4 and its subplots:**
>
> In the **Visualization Analysis** section, we mentioned that the curves represent "probability distributions for the four edges (left, top, right, bottom)." These subplots illustrate how D-FINE adjusts each side of the bounding box based on the model's confidence. The unweighted distributions reflect the model's confidence in applying various levels of adjustment to each edge, while the weighted distributions show the actual adjustments made.
>
> Here's what the different elements represent:
>
> - **Subplots (Left, Top, Right, Bottom):** Each subplot corresponds to one edge of the bounding box and shows how adjustments are made for that specific edge.
> - **Peaks and Shifts:** When the initial bounding box is sufficiently accurate, the unweighted distribution tends to be symmetric and centered near 16, leading to a weighted adjustment close to 0 (i.e., no adjustment needed). Peaks shifting to the left or right indicate that the edge should contract inward or expand outward, respectively. Peaks near the extremes of the distribution represent more significant adjustments.
> - **Colored Lines:** The red and green lines represent these boxes and distributions belonging to the first and last layers.
>
> We will revise the manuscript to include a more detailed explanation of these plots, along with an expanded caption for Figure 4, to clarify their purpose and interpretation. Thank you for bringing this to our attention.
>
> ---
>
> Thank you again for your valuable feedback. We believe that these clarifications and revisions strengthen our paper, and we appreciate your help in improving its quality.

---

> ### Comment · Area_Chair_Q53A · 2024-11-25
> **review the rebuttal**
>
> Dear Reviewer oAYL,
>
> Could you kindly review the rebuttal thoroughly and let us know whether the authors have adequately addressed the issues raised or if you have any further questions.
>
> Best,
>
> AC of Submission1611

---

> ### Comment · Reviewer_oAYL · 2024-11-25
>
> Thanks to the authors for the rebuttal and responding to the comments.
> I am still not comfortable with Figure 4. Indeed, bbox edges are not independent and they are influenced by each other during the refinement process. So, separately visualising them misrepresents the interdependencies and shows an incomplete picture of the bbox. Moreover, subplots for each edge lack coherence in showing how the whole bbox improves during refinement. Maybe using confidence distributions for the whole bbox is a better choice.
> Further, Figure 3 is difficult to understand. Additional details in the caption can improve comprehension.

---

> ### Author Response · Authors · 2024-11-25
>
> Thank you for your feedback and for pointing out your concerns, we address the main concerns as follows.
>
> **Regarding Figure 4 and Independent Modeling of Bounding Box Edges:**
>
> We agree that bounding box edges are influenced by each other during the refinement process. **However**, one of the key motivations of our work, as stated in **Introduction (L062-L065)** and **Method (L188-L190)**, is to independently model each bounding box edge. This design allows us to explicitly capture the uncertainties and refinements needed for individual edges.
>
> 1. **Dependent or Independent on Bounding Box Edges:**
> You may have misunderstood the process of edge refinement. Each edge does not require determining two endpoints. Instead, D-FINE independently adjusts the distance of each edge relative to the center of the bounding box. This concept is defined in **PRELIMINARIES (L138-L155)** and **Method (L206-L227)**, where the bounding box distances are explained.
>
> 2. **Rationale Behind current Edge_Specific Visualizations:**
>    Figure 4 was designed to highlight the edge-specific adjustments made by D-FINE. In many cases, initial bounding box predictions are already accurate for some edges, requiring no significant adjustments, while other challenging edges with larger errors undergo targeted refinements. By visualizing the probability distributions of individual edges, we demonstrate how FDR effectively prioritizes the necessary corrections without introducing unnecessary changes.
>
> 3. **Concerns on Whole-Box Visualization:**
>    While whole-box confidence distributions may provide a valuable global perspective, they do not align with the core functionality of FDR, which is designed to independently refine each edge. Such visualizations may overlook the fine-grained adjustments and edge-specific optimizations that are central to our approach.
>
> **Regarding Figure 3:**
>
> Thank you for pointing out your concerns. We agree that additional clarification can improve comprehension.
> We will revise the caption to include more details about the elements and interactions depicted in the figure.
>
> The **updated caption** is as follows:
>
> **Figure 3:** Overview of the GO-LSD process. Localization knowledge from the final layer (teacher) is distilled into shallower layers (students) through DDF loss with decoupled weighting strategies. Orange and green curves represent the probability distributions of unmatched and matched predictions, respectively, with decoupled weights $\alpha_k$ and $\beta_k$ controlling their contributions. Arrows indicate the flow of self-distillation, showing how refined distributions at the final layer guide shallower layers’ optimization.
>
> These updates will be reflected in the revised manuscript after reformatting. Thank you for your valuable suggestion, which helps to improve the clarity and accessibility of our work.

---

> ### Author Response · Authors · 2024-12-02
> **Looking forward to hearing from you**
>
> Dear Reviewer **oAYL**,
>
> We sincerely appreciate your time and effort in reviewing our manuscript and providing valuable feedback. Below is a summary of how we have addressed your concerns:
>
> 1. **Fixed Coordinates and Localization Uncertainty**
>    - We clarified that our approach complements, rather than undermines, fixed-coordinate methods. We acknowledge the success of methods like Faster R-CNN and RetinaNet and emphasize that our work seeks to enhance localization uncertainty modeling, building on existing advancements.
>
> 2. **Refining Smaller Residuals**
>    - We provided a clearer explanation of "refining smaller residuals," highlighting that as early layers make more accurate bounding box predictions, subsequent layers focus on fine-tuning smaller residuals rather than correcting large errors.
>
> 3. **Softmax vs. Other Normalization Methods (e.g., Entmax)**
>    - We explained our choice of Softmax over alternatives like Entmax, citing its consistency with KL divergence and the need to preserve small probabilities. We also discussed the computational efficiency of Softmax, particularly in real-time detection tasks, where it outperforms Entmax.
>
> 4. **Figure 4 and Edge-Specific Refinement**
>    - We clarified that Figure 4 visualizes the independent adjustment of each bounding box edge. This approach helps capture edge-specific uncertainties and ensures targeted refinements. We also explained that visualizing whole-box confidence distributions may not align with the core functionality of our method, which focuses on edge-specific adjustments.
>
> 5. **Figure 3 and Caption Update**
>    - We agreed that Figure 3 needed further clarification and revised its caption to include more details about the GO-LSD process. The updated caption now provides a clearer explanation of the interactions between matched and unmatched predictions, as well as the flow of self-distillation.
>
> We hope that these revisions adequately address your concerns. Please let us know if there are any remaining issues or further questions. We look forward to your feedback.
>
> Best regards,
> D-FINE Authors

---

> > ### Comment · Reviewer_oAYL · 2024-12-02
> >
> > Thanks for the explanations. I increased my score.

---

> > > ### Author Response · Authors · 2024-12-03
> > >
> > > Dear Reviewer **oAYL**,
> > >
> > > We are delighted to hear your positive feedback!
> > >
> > > The **Reviewer-Author** discussion session is drawing to an end. We greatly value the opportunity to discuss with you and welcome any additional feedback or question you may have.
> > >
> > > Please let us know if there are any other aspects we should look into to further improve this work.
> > >
> > > Best regards,
> > >
> > > D-FINE Authors

---

### Author Response · Authors · 2024-11-19

We thank all reviewers for their detailed reviews and constructive feedback. We have made the following changes to the paper and uploaded a new revision:

- **Improved Explanations and Visualizations:** We have redrawn all figures for better clarity and provided clearer explanations of key concepts.

- **Language Refinements and Consistency:** Revised ambiguous phrases for better readability, ensured consistency in variable names throughout the manuscript, and clarified our claims about fixed-coordinate regression methods to acknowledge their success and explain how our approach complements existing methods.

- **Extended Experiments:** Added evaluations on the CrowdHuman and Objects365 datasets to demonstrate the generalizability and robustness of D-FINE across different object detection contexts.

Overall, we are pleased that the reviewers found our method innovative and well-presented. Your feedback has been invaluable in improving our paper, and we believe these revisions strengthen our work.

---

### Meta-Review · Area_Chair_Q53A · 2024-12-14

**Metareview:**

(a) The paper presents D-FINE, a real-time object detection model improving DETR. It introduces Fine-grained Distribution Refinement (FDR) for refined bounding box distributions and Global Optimal Localization Self-Distillation (GO-LSD) to enhance earlier predictions. Experiments on COCO show D-FINE is fast and competitive with current detectors.

(b) The paper is clear and well-structured, supported by thorough experiments on COCO and Objects365. D-FINE outperforms many real-time detectors, balancing speed and accuracy effectively. Ablation studies and hyperparameter tuning provide valuable insights. Comprehensive comparisons with DETR-based models and real-world visualizations validate its robustness.

(c) The main weaknesses of the first version included insufficient explanations and visualizations, ambiguous phrasing, inconsistent variable names, and a lack of experiments on additional object detection datasets. These issues were all resolved during the rebuttal.

(d) The primary reason for acceptance is the solid, practical performance of the model for real-time detection and its innovative approaches, including distribution refinement and localization self-distillation. This work has the potential to significantly impact the object detection community.

**Additional Comments On Reviewer Discussion:**

(1) Reviewer oAYL raises concerns about the claim that fixed-coordinate regression inherently limits performance, citing successes like Faster R-CNN and RetinaNet, and suggests clarifying this assertion while highlighting the proposed method's advantages. They seek clarification on the phrase "refining smaller residuals." Concerns are also noted about Eq. (3), where softmax normalization may hinder learning in deeper layers and reduce localization precision, suggesting alternatives like Entmax normalization. For Figure 4, the reviewer points out a lack of explanation for subplots and colored lines, requesting detailed captions and clarification of their significance. The authors address the main issues, and Reviewer oAYL raises additional questions about the clarity of Figures 3 and 4. The authors provide further details about the mechanisms depicted in the figures, which finally resolve all concerns.

(2) Reviewer hveE finds the term "layer" ambiguous in the contexts of FDR and GO-LSD, seeking clarification on its meaning. They also request an explicit explanation of the gray squares in Figure 3, suggesting adding this information to the caption or legend. The authors successfully address all concerns in one-shot response.

(3) Reviewer 5Uux suggests testing D-FINE on diverse datasets like CrowdHuman and LVIS could validate its generalizability and robustness in different detection scenarios. The authors show more results on CrowdHuman and Object365, two popular object detection benchmarks.

(4) Reviewer 759J points out inconsistencies in performance reporting and variable names, as well as the need for language refinement to avoid ambiguity. The authors have effectively addressed these issues.

---

### Decision · Program_Chairs · 2025-01-22

Accept (Spotlight)